# Endogenous oligomer formation underlies DVL2 condensates and promotes Wnt/β-catenin signaling

Senem Ntourmas[1], Martin Sachs[1], Petra Paclíková[2], Martina Brückner[1], Vítězslav Bryja[2], Jürgen Behrens[1], Dominic B Bernkopf[1]*

[1]Experimental Medicine II, Nikolaus-Fiebiger-Center, Friedrich-Alexander University Erlangen-Nürnberg, Erlangen, Germany; [2]Department of Experimental Biology, Faculty of Science, Masaryk University, Brno, Czech Republic

## eLife Assessment

This **valuable** study contributes to the understanding of phase separation in Dishevelled (DVL) proteins, by investigating the endogenous complexes of DVL2 using ultracentrifugation and contrasting them with DVL1 and DVL3 behaviour and the functional validation of the DVL2 intrinsically disordered regions mediating the protein condensate. The study includes a **solid** characterisation of several overexpression constructs, including in KO cells. However, investigations of the roles of the described DVL2 regions at the endogenous level remain to be carried out.

*For correspondence: dominic.bernkopf@fau.de

**Abstract** Activation of the Wnt/β-catenin pathway crucially depends on the polymerization of dishevelled 2 (DVL2) into biomolecular condensates. However, given the low affinity of known DVL2 self-interaction sites and its low cellular concentration, it is unclear how polymers can form. Here, we detect oligomeric DVL2 complexes at endogenous protein levels in human cell lines, using a biochemical ultracentrifugation assay. We identify a low-complexity region (LCR4) in the C-terminus whose deletion and fusion decreased and increased the complexes, respectively. Notably, LCR4-induced complexes correlated with the formation of microscopically visible multimeric condensates. Adjacent to LCR4, we mapped a conserved domain (CD2) promoting condensates only. Molecularly, LCR4 and CD2 mediated DVL2 self-interaction via aggregating residues and phenylalanine stickers, respectively. Point mutations inactivating these interaction sites impaired Wnt pathway activation by DVL2. Our study discovers DVL2 complexes with functional importance for Wnt/β-catenin signaling. Moreover, we provide evidence that DVL2 condensates form in two steps by pre-oligomerization via high-affinity interaction sites, such as LCR4, and subsequent condensation via low-affinity interaction sites, such as CD2.

## Introduction

The Wnt/β-catenin signaling pathway promotes the proliferation of stem cells, orchestrating self-renewal and regeneration of epithelial tissues in adults (*Clevers et al., 2014*). Deregulation of the pathway has been causally associated with severe pathologies, most prominently colorectal cancer (*Clevers and Nusse, 2012*). In the absence of Wnt ligands, β-catenin-dependent Wnt signaling is continuously silenced via glycogen synthase kinase 3 beta (GSK3B)-mediated phosphorylation targeting β-catenin for proteasomal degradation (*Stamos and Weis, 2013*). Phosphorylation of β-catenin is induced by the scaffold protein AXIN1, which interacts with β-catenin and GSK3B (*Stamos and Weis, 2013*). Upon binding of Wnt ligands to frizzled receptors and low-density lipoprotein

receptor-related protein 5 or 6 (LRP5/6) co-receptors, the positive Wnt pathway regulator dishevelled 2 (DVL2) interacts with frizzled and clusters beneath the plasma membrane (*MacDonald and He, 2012*). DVL2 clusters recruit AXIN1 and GSK3B from the cytosol, and together these proteins assemble sphere-like signalosomes at the membrane (*Bilic et al., 2007*). Within these signalosomes, GSK3B activity is redirected from β-catenin to LRP5/6 and finally inhibited (*Bilic et al., 2007*; *Taelman et al., 2010*). In consequence, β-catenin accumulates and can translocate into the nucleus, where it promotes transcription of its target genes (*Behrens et al., 1996*).

Activation of Wnt/β-catenin signaling by DVL2 crucially depends on DVL2 polymerization via its N-terminal DIX domain (*Kishida et al., 1999*; *Schwarz-Romond et al., 2007a*). However, the low auto-affinity of the DIX domain in the mid-micromolar range and the low cellular concentration of DVL2 strongly disfavor polymerization, and only the pre-clustering of DVL2 at Wnt-receptor-complexes is suggested to overcome this problem (*Bienz, 2014*). Mechanistically, DVL2 polymerization may support DVL2 clustering, AXIN1 recruitment, and/or signalosome formation (*Bilic et al., 2007*; *Schwarz-Romond et al., 2007a*; *Schwarz-Romond et al., 2007b*). When DVL2 is overexpressed, DIX-mediated polymerization gives rise to microscopically visible DVL2 assemblies, which are sphere-like, membrane-free, and highly dynamic (*Schwarz-Romond et al., 2005*). Recently, these DVL2 assemblies were characterized as phase-separated biomolecular condensates (*Kang et al., 2022*), which represent a functionally diverse class of membrane-free cell organelles with common biophysical properties (*Banani et al., 2017*; *Shin and Brangwynne, 2017*). In addition to the DIX domain, other parts of the protein contribute to DVL2 condensation and activity, such as the DEP domain or an intrinsically disordered region (*Gammons et al., 2016*; *Kang et al., 2022*; *Vamadevan et al., 2022*), suggesting that major regions of DVL2 are evolutionarily optimized for condensation. Moreover, condensation of DVL2 appears to be a regulated process controlled through posttranslational modification, such as ubiquitination, and depending on its conformation, open versus close form (*Lee et al., 2015*; *Vamadevan et al., 2022*). However, there is still a controversial debate on whether DVL2 forms condensates at endogenous expression levels, as recent studies report only small DVL2 assemblies with less than 10 molecules (*Kan et al., 2020*), or only one big DVL2 condensate at the centrosome (*Schubert et al., 2022*), or about 100 condensates per cell with sizes of 0.2–0.5 μm (*Kang et al., 2022*).

In vertebrates, three DVL paralogs exist, DVL1, DVL2, and DVL3. Although overexpression of each paralog activates Wnt/β-catenin signaling, loss-of-function studies consistently report that DVL1, DVL2, and DVL3 exhibit different capabilities to transduce Wnt signals and that overexpression of one paralog does not compensate for the loss of another (*Lee et al., 2008*; *Paclíková et al., 2021*). Thus, these studies point to non-redundant molecular functions of the DVL paralogs, yet, their molecular differences remain poorly understood.

Here, we report biochemical evidence for endogenous DVL2 complexes consisting of at least eight molecules, supporting the idea of DVL2 polymerization at endogenous expression levels. Using DVL2 deletion and point mutants, we mapped and characterized a low-complexity region in the DVL2 C-terminus that promoted complex formation through mediating intermolecular DVL2 self-interaction. Our data suggest that these complexes most likely represent underlying substructures of DVL2 biomolecular condensates, which precede and initiate condensation. Moreover, the discovered oligomeric DVL2 complexes were of functional importance because point mutations that impaired complex formation attenuated Wnt pathway activation by DVL2.

## Results

### Endogenous DVL2 forms oligomeric complexes

Performing ultracentrifugation assays, endogenous DVL2 (79 kDa) penetrated far deeper into a sucrose density gradient than AXIN1 (96 kDa) in spite of its lower molecular weight, indicating that DVL2 forms protein complexes (*Figure 1A and E*). Noteworthy, most DVL2 molecules appeared to be engaged in these complexes (*Figure 1A*). The complexes occurred in different cell lines (*Figure 1—figure supplement 1A*), and were detectable with two, siRNA-validated antibodies (*Figure 1—figure supplement 1B, C*). DVL2 (79 kDa) showed a fractionation pattern similar to thyroglobulin (669 kDa), a commercial molecular weight marker, suggesting DVL2 complex sizes of about eight molecules assuming homotypic complexes (*Figure 1A and B*). As a control, AXIN1 (96 kDa) showed a fractionation pattern

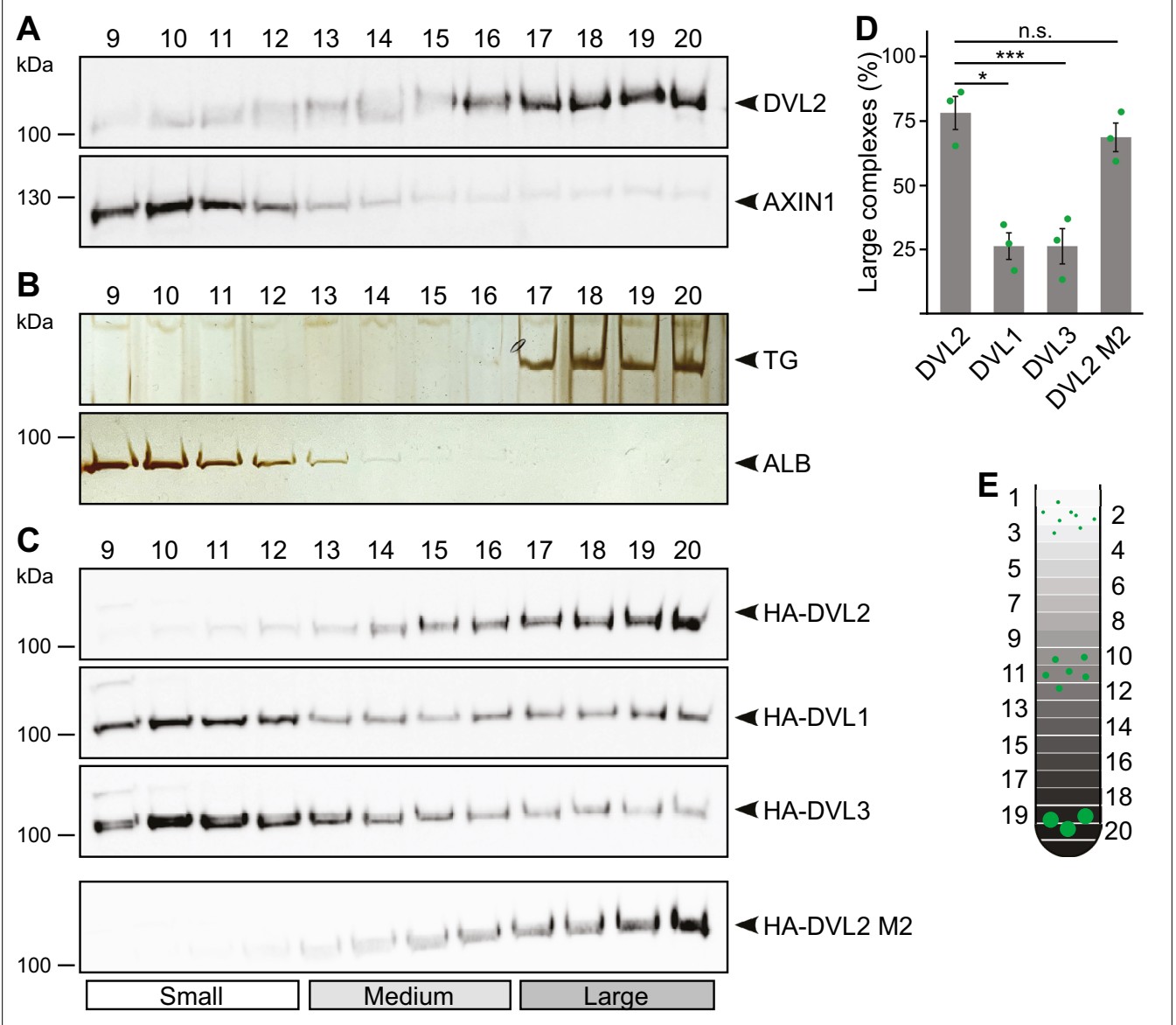

**Figure 1.** Endogenous dishevelled 2 (DVL2) forms paralog-specific oligomeric complexes. (**A,C**) Western blotting for indicated endogenous (**A**) or transiently expressed proteins, detected with anit-HA antibodies (**C**) after fractionation of HEK293T cell lysates via sucrose density ultracentrifugation. (**B**) Silver staining of thyroglobulin (TG) or albumin (ALB) after sucrose density ultracentrifugation of purified proteins. (**A-C**) shows one out of at least three representative experiments. Analyzed fractions are indicated above the blots according to (**E**). (**D**) Amount of the protein that was engaged in large complexes (see label Large in **C**), relative to the cumulative protein amount detected in all investigated fractions, as determined by 2D densitometry analysis of protein bands from three independent experiments as in (**C**) (n=3). Results are mean ± SEM, *p<0.05, ***p<0.001 (Student's *t*-test). (**E**) Schematic representation of the ultracentrifugation assay, illustrating the distribution of proteins of different sizes (green) within numbered fractions of a sucrose gradient form low (light gray) to high density (black).

The online version of this article includes the following source data and figure supplement(s) for figure 1:

**Source data 1.** Excel file providing the numerical source data to *Figure 1*.

**Source data 2.** PDF files containing the original, labeled blots and gels to *Figure 1*.

**Source data 3.** TIF files of the raw blots and gels to *Figure 1*.

**Figure supplement 1.** Endogenous dishevelled 2 (DVL2) forms complexes.

**Figure supplement 1—source data 1.** PDF files containing the original, labeled blots to *Figure 1—figure supplement 1*.

**Figure supplement 1—source data 2.** TIF files of the raw blots to *Figure 1—figure supplement 1*.

similar to albumin (66 kDa) suggesting monomeric precipitation (*Figure 1A and B*). Interestingly, the DVL2 complexes appeared to be paralog-specific because complexes were almost absent for DVL1 and DVL3 (*Figure 1C and D*). The persistence of the complexes at low protein concentrations in cellular extracts indicated that they form via interaction sites with rather high affinity. Although the DIX domain is the best-characterized polymerization domain in DVL2 (*Bienz, 2014*), its low auto-affinity suggests that it is not involved in the formation of these complexes (*Schwarz-Romond et al., 2007a*). The striking difference between DVL2 and AXIN1 pointed in the same direction (*Figure 1A*), since both proteins contain a functional DIX domain (*Kishida et al., 1999*). Consistently, a DIX domain-inhibiting point mutation (DVL2 M2) (*Schwarz-Romond et al., 2007a*) did not affect DVL2 complexes (*Figure 1C and D*).

## A low-complexity region in the C-terminus promotes DVL2 complexes

Deletion of the DEP domain (construct ΔDEP) decreased DVL2 complexes, in line with its reported function in DVL2-DVL2 interaction (*Gammons et al., 2016*). However, additional deletion of the remaining C-terminus (construct 1–418) markedly reduced them further, demonstrating a strong contribution of the deleted residues 521–736 to complex formation (*Figure 2A–C*). Importantly, decreased protein complexation of DVL2 ΔDEP and 1–418 in ultracentrifugation experiments conspicuously correlated with decreased formation of condensates in immunofluorescence-based assays (*Figure 2D and E*) and decreased Wnt pathway activation in reporter assays (*Figure 2F*). Therefore, we hypothesized that the DVL2 complexes may be important for signaling activity. To identify candidate regions for a more precise mapping within residues 521–736, we used the SEG algorithm predicting low-complexity regions (*Wootton and Federhen, 1993*), the TANGO algorithm predicting aggregation (*Fernandez-Escamilla et al., 2004*) and protein alignments (*Sievers et al., 2011*). We identified four low-complexity regions (LCR1-4), which are associated with protein assembly (*Martin et al., 2020*; *Martin and Mittag, 2018*), one potential aggregation site embedded in LCR4, and two domains whose evolutionary conservation may point to functional importance (CD1-2, *Figure 1—figure supplement 1D*). Since the deletion of residues 521–736 showed strong effects when combined with the DEP deletion (1–418 vs ΔDEP, *Figure 2*), we performed the following mapping in the ΔDEP context. Given the good correlation between complexes and condensates (*Figure 2C and E*), we decided to use immunofluorescence-based analysis of condensates for mapping, as it is more convenient than density gradient ultracentrifugation. Upon individual deletion of the six identified regions LCR1-4 and CD1-2, only deletion of LCR4 and CD2 decreased condensate formation and Wnt pathway activation of DVL2 ΔDEP (*Figure 3A*; *Figure 3—figure supplement 1A–C*). Combined deletion of LCR4 and CD2 increased the effect (*Figure 3A*; *Figure 3—figure supplement 1A–C*). We, therefore, consider the two adjacent regions as one functional unit, hereafter referred to as condensate forming region (CFR). DVL2 ΔDEP-ΔCFR exhibited a marked decrease in the number of cells with condensates (*Figure 3A–C*), in the number of condensates per cell (*Figure 3D*; *Figure 3—figure supplement 1D–G*) and in Wnt pathway activation (*Figure 3E*), similar to 1–418. Importantly, individual or combined fusion of LCR4 and CD2 to 1–418 sufficed to induce condensation (*Figure 3A–D*; *Figure 3—figure supplements 1D-G and Figure 3—figure supplements 2A and B*) and Wnt pathway activation (*Figure 3E*; *Figure 3—figure supplement 2C and D*), rendering 1–418+CFR as active as ΔDEP. Mutational inactivation of the DIX domain (1–418+CFR M2) abolished condensates demonstrating that the DIX domain is required for CFR-mediated condensates (*Figure 3—figure supplement 2A and B*). Interestingly, the investigated DVL2 mutant proteins predominantly formed nuclear condensates in contrast to the cytosolic condensates of WT DVL2, most likely, because a nuclear export signal (*Figure 2A*) was deleted in these mutants (*Figure 3A*). However, investigating only cells with cytosolic condensates (*Figure 3—figure supplement 2E and F*) revealed similar differences between the DVL2 mutants as were observed when investigating mainly cells with nuclear condensates (*Figure 3C*; *Figure 3—figure supplement 2B*), suggesting that the detected differences are not due to nuclear localization but reflect the overall condensation capacity of the DVL2 mutants. Moreover, fusion of CFR to the isolated DVL2 DIX or AXIN1 DAX domain sufficed to trigger condensation, which was prevented by M2/M3 mutation of the DIX/DAX domain (*Figure 3—figure supplement 2D,G and H*). Thus, loss- and gain-of-function experiments identified CFR as the crucial region for condensation and Wnt pathway activation within the DVL2 C-Terminus, which functionally cooperates with the DIX domain.

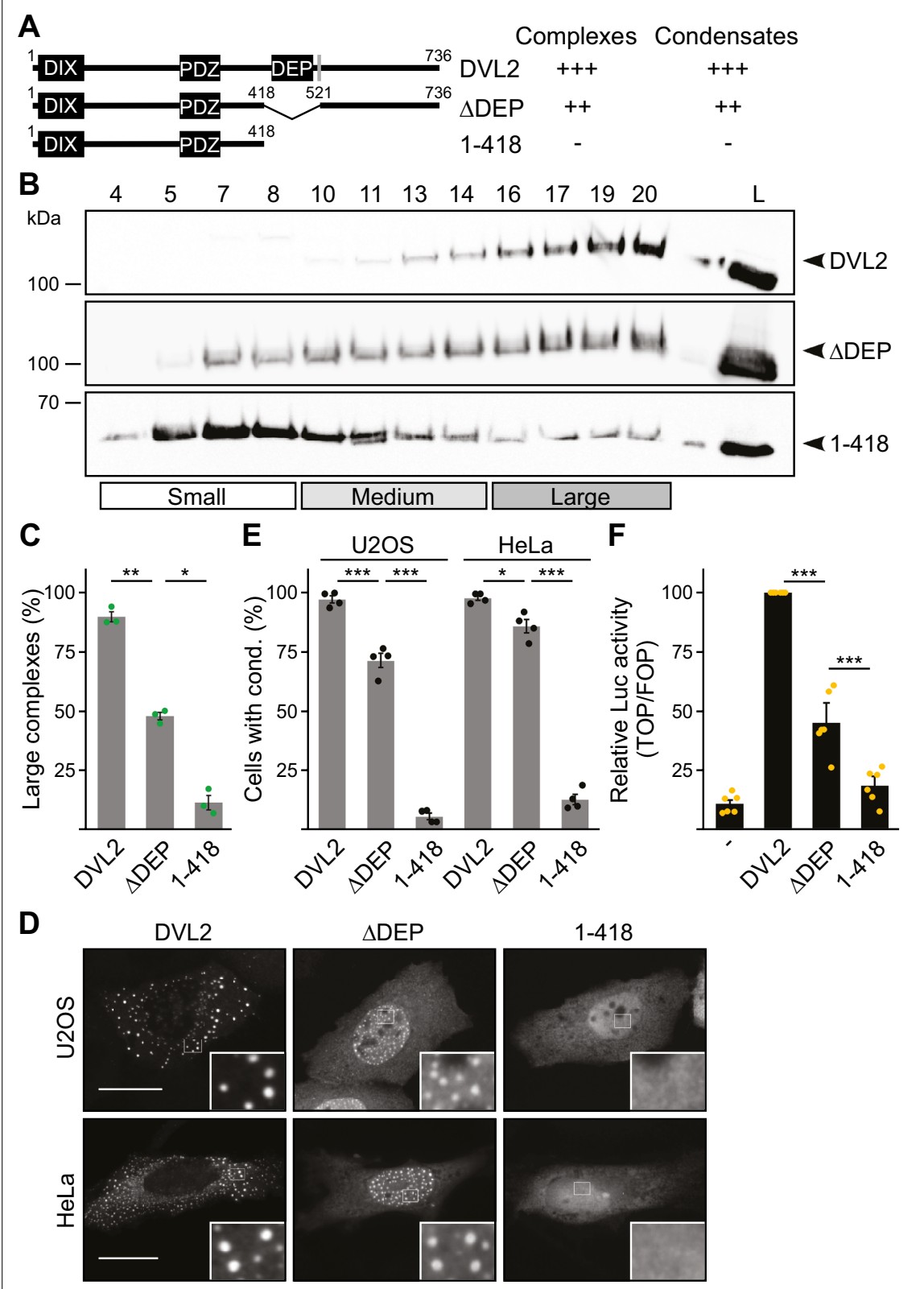

**Figure 2.** The dishevelled 2 (DVL2) C-terminus promotes complexes, condensates, and activity. (**A**) To scale schemes of DVL2 constructs with the DIX, the PDZ, and the DEP domain. A nuclear export signal is highlighted in gray (*Itoh et al., 2005*). Indicated complexation and condensation summarizes the findings in (**B–E**). (**B**) Western blotting for indicated transiently expressed proteins bevor (L) and after fractionation (4-20) of HEK293T cell lysates via sucrose density ultracentrifugation. (**C**) Percentage of the protein that was engaged in large complexes as specified in (**B**) (n=3, refer to the legend

*Figure 2 continued on next page*

*Figure 2 continued*

in *Figure 1D* for more details). (**D**) Immunofluorescence of indicated HA-tagged proteins in transiently transfected U2OS and HeLa cells. Scale bars: 20 μm. Insets are magnifications of the boxed areas. Interestingly, DEP domain deleted constructs frequently showed nuclear condensates in contrast to the cytosolic condensates of full length DVL2, which is most likely explained by the deletion of a nearby nuclear export signal (see **A**) and which still allowed determining differences in condensation capacity. (**E**) Percentage of cells with condensates out of 1200 transfected cells from four independent experiments as in (**D**) (n=4). (**F**) Relative luciferase activity reporting β-catenin-dependent transcription in HEK293T cells expressing the indicated constructs (n=6). (**C, E, F**) Results are mean ± SEM, *p<0.05, **p<0.01, ***p<0.001 (Student's *t*-test).

The online version of this article includes the following source data for figure 2:

**Source data 1.** Excel file providing the numerical source data to *Figure 2*.

**Source data 2.** PDF file containing the original, labeled blots to *Figure 2*.

**Source data 3.** TIF files of the raw blots to *Figure 2*.

Next, we investigated whether CFR contributes to the formation of DVL2 complexes detected by density gradient ultracentrifugation. Indeed, CFR deletion from ΔDEP (ΔDEP-ΔCFR) and fusion with 1–418 (1–418+CFR) decreased and increased complexes, respectively (*Figure 4A–D*). Moreover, replacing the respective sequence in DVL1 with the CFR of DVL2 (DVL1-CFR^DVL2) promoted complex formation (*Figure 4E and F*). Surprisingly, only LCR4 but not CD2 was required for complex formation when deleted from ΔDEP or mediated complex formation when fused to 1–418 (*Figure 4A–D*), although both parts were required for condensate formation (*Figure 3—figure supplement 1B*). Of note, LCR4 is not well conserved in DVL1 and DVL3, which do not form complexes (*Figure 1C*; *Figure 1—figure supplement 1D*). Together, our data revealed a bipartite, 58 amino acid region at the very C-Terminus of DVL2 consisting of a low complexity region (LCR4) that promotes complexes and condensates and a conserved domain (CD2) that only promotes condensates and is dispensable for complexes.

## LCR4 and CD2 cooperatively mediate DVL2 self-interaction

Given the role of the CFR parts LCR4 and CD2 in complex and condensate formation of DVL2, we speculated that they might directly mediate DVL2-DVL2 interaction. To analyze the role of a putative aggregation site (aggregon) that was predicted by the TANGO algorithm within LCR4 (*Figure 1—figure supplement 1D*), we designed mutations of two key valine residues (VV-AA) predicted to prevent aggregation (*Figure 5A*; *Figure 5—figure supplement 1A*). LCR4 VV-AA mutation of DVL2 ΔDEP decreased condensate formation as efficiently as LCR4 deletion (*Figure 5B and C*), and fusion of VV-AA-mutated LCR4 failed to increase condensate formation of 1–418, in contrast to WT LCR4 (*Figure 5—figure supplement 1B and C*). In CD2, the TANGO algorithm did not predict aggregation sites. However, we detected interspersed phenylalanine residues in CD2 (*Figure 5A*), which might promote interaction through stacking of their aromatic rings, acting as 'stickers' of unstructured regions, as recently described (*Martin et al., 2020*). CD2 FF-AA mutation decreased condensate formation and Wnt pathway activation as efficiently as CD2 deletion (*Figure 5B and C*; *Figure 5—figure supplement 1D*), and CD2 FF-AA fusion failed to increase condensate formation or Wnt pathway activation, in contrast to WT CD2 (*Figure 5—figure supplement 1E–G*). Our data suggest that CFR is a bipartite protein interaction site for self-association of DVL2. In line with this, the isolated CFR, which exhibited a homogeneous cellular distribution when expressed alone, accumulated within DVL2 condensates upon co-expression of both proteins (*Figure 5D and E*), suggesting CFR-DVL2 interaction. CFR VV-AA FF-AA mutation markedly reduced this CFR-DVL2 co-localization (*Figure 5D and E*). Moreover, DVL2 CFR did not co-localize with DVL1 or DVL3 condensates (*Figure 5—figure supplement 1H*), consistent with the low conservation of the LCR4 aggregon in these paralogs (*Figure 1—figure supplement 1D*). Notably, co-expression of the isolated CFR inhibited Wnt pathway activation by DVL2 in a dosage-dependent manner, which was attenuated by CFR FF-AA mutation (*Figure 5F*). In this experiment, free CFR might weaken DVL2-DVL2 interaction by saturating the interaction surface of the CFR within DVL2. Together, our data provide evidence that CFR mediates intermolecular DVL2 self-interaction via an aggregation site in LCR4 and phenylalanine stickers in CD2. Since individual deletion of LCR4 or CD2 was sufficient to impair condensates (*Figure 3—figure supplement 1B*), both interaction sites most likely cooperate to drive the condensate formation of DVL2.

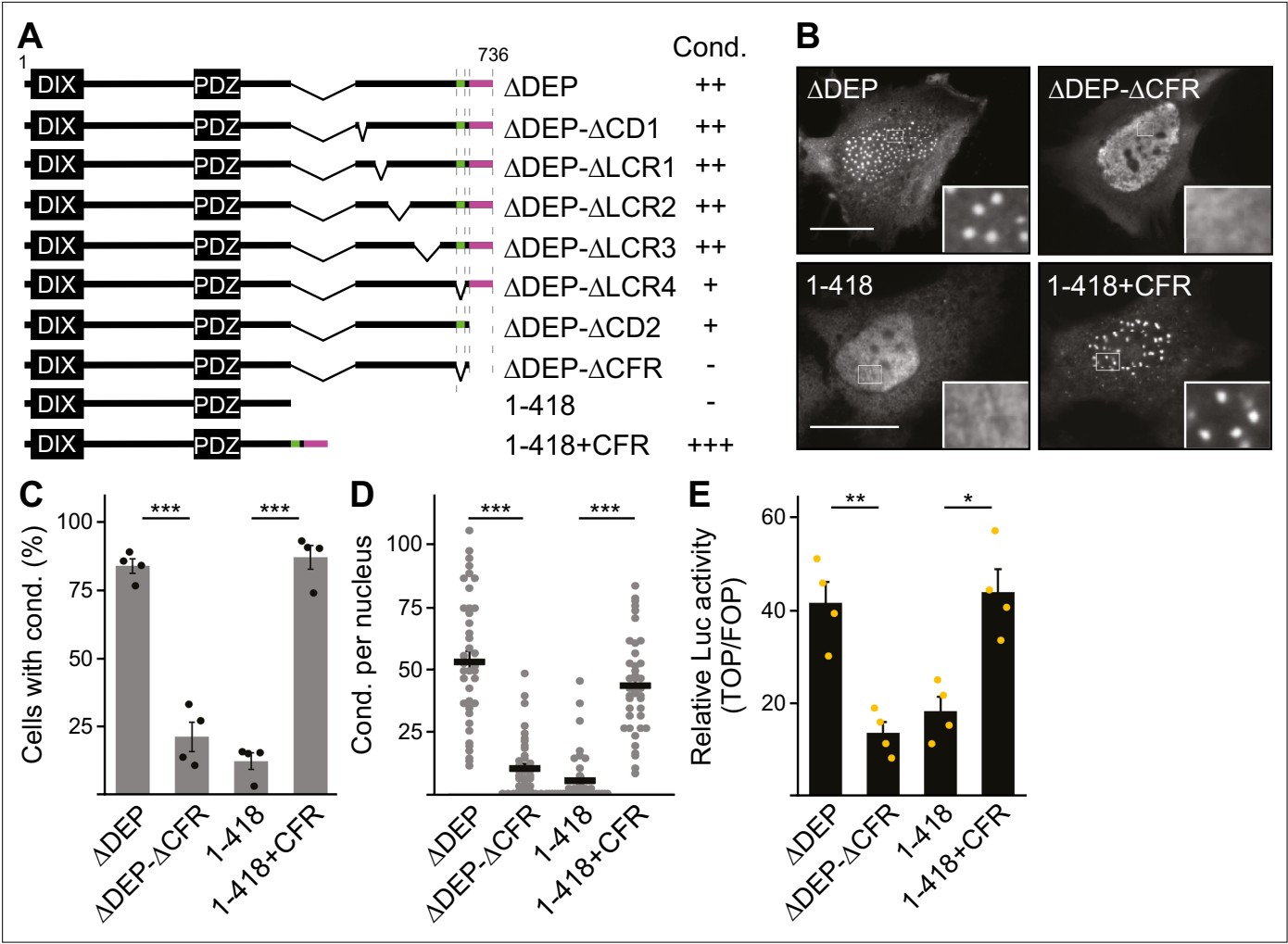

**Figure 3.** A 58 aa C-terminal region promotes dishevelled 2 (DVL2) condensates and activity. (**A**) To scale schemes of DVL2 constructs. Indicated condensation (Cond.) summarizes the findings in *Figure 3—figure supplement 1B*, and the identified crucial regions are highlighted in green (low-complexity region, LCR4) and magenta (conserved domain, CD2). (**B**) Immunofluorescence of indicated HA-tagged proteins in transiently transfected HeLa cells. Scale bars: 20 μm. Insets are magnifications of the boxed areas. (**C**) Percentage of cells with condensates out of 1200 transfected cells from four independent experiments as in (**B**) (n=4). (**D**) Automated quantification of condensate number per nucleus by the Icy Spot Detector (*Olivo-Marin, 2002*) in 40 cells from four independent experiments as in (**B**) (n=40). (**E**) Relative luciferase activity reporting β-catenin-dependent transcription in HEK293T cells expressing the indicated constructs (n=4). (**C–E**), Results are mean ± SEM, *p<0.05, **p<0.01, ***p<0.001 (Student's *t*-test).

The online version of this article includes the following source data and figure supplement(s) for figure 3:

**Source data 1.** Excel file providing the numerical source data to *Figure 3*.

**Figure supplement 1.** Mapping of dishevelled 2 (DVL2) regions promoting condensates and activity.

**Figure supplement 1—source data 1.** Excel file providing the numerical source data to *Figure 3—figure supplement 1*.

**Figure supplement 2.** Low-complexity region (LCR4) and conserved domain (CD2) cooperate to promote DIX domain-dependent condensates.

**Figure supplement 2—source data 1.** Excel file providing the numerical source data to *Figure 3—figure supplement 2*.

## DVL2 CFR promotes phase separation

While 1–418 and DVL2 DIX exhibited a homogenous cellular distribution, fusion of CFR-induced spherical, microscopically visible condensates of 1–418+CFR and DIX+CFR (*Figure 3B*; *Figure 3—figure supplement 2G*). To study the nature of CFR-mediated condensates, we treated cells expressing DVL2, 1–418+CFR, or DIX+CFR with a hypoosmolar buffer (osmotic shock) or with the bivalent alcohol 1,6-hexanediol, as previously done in biomolecular condensate research to challenge phase separation (*Nott et al., 2015*; *Ribbeck and Görlich, 2002*). Both treatments significantly decreased condensates of all three studied proteins within 3 min for osmotic shock and 1 hr for 1,6-hexanediol (*Figure 6A–C*;

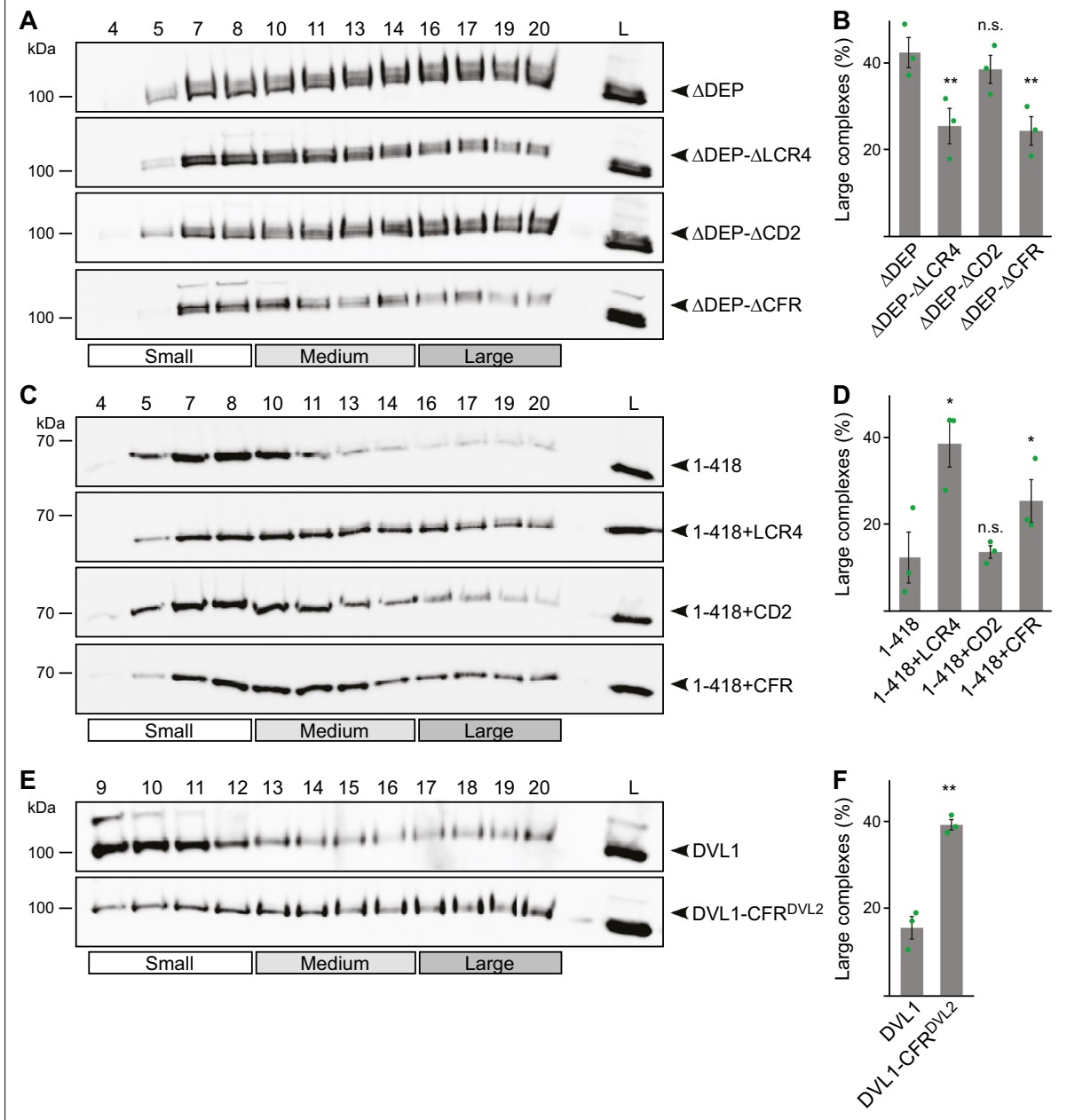

**Figure 4.** Low-complexity region (LCR4) mediates complex formation. (**A, C, E**) Western blotting for indicated transiently expressed proteins bevor (**L**) and after fractionation (4-20) of HEK293T cell lysates via sucrose density ultracentrifugation. (**B, D, F**) Percentage of the protein that was engaged in large complexes as specified in **A, C** or **E** (n=3, refer to the legend to *Figure 1D* for more details). Results are mean ± SEM, *p<0.05, **p<0.01 (Student's *t*-test).

The online version of this article includes the following source data for figure 4:

**Source data 1.** Excel file providing the numerical source data to *Figure 4*.

**Source data 2.** PDF files containing the original, labeled blots to *Figure 4*.

**Source data 3.** TIF files of the raw blots to *Figure 4*.

*Videos 1 and 2*). These findings are consistent with the transition from a two-phase state of condensates with high protein concentration and surrounding spaces with low protein concentration to a one-phase state of homogenous protein distribution (*Figure 6A*). We concluded that CFR indeed induced phase separation to promote 1–418+CFR and DIX+CFR condensates, in line with the fact that WT DVL2 was shown to undergo phase separation (*Kang et al., 2022*). Moreover, the fusion of an

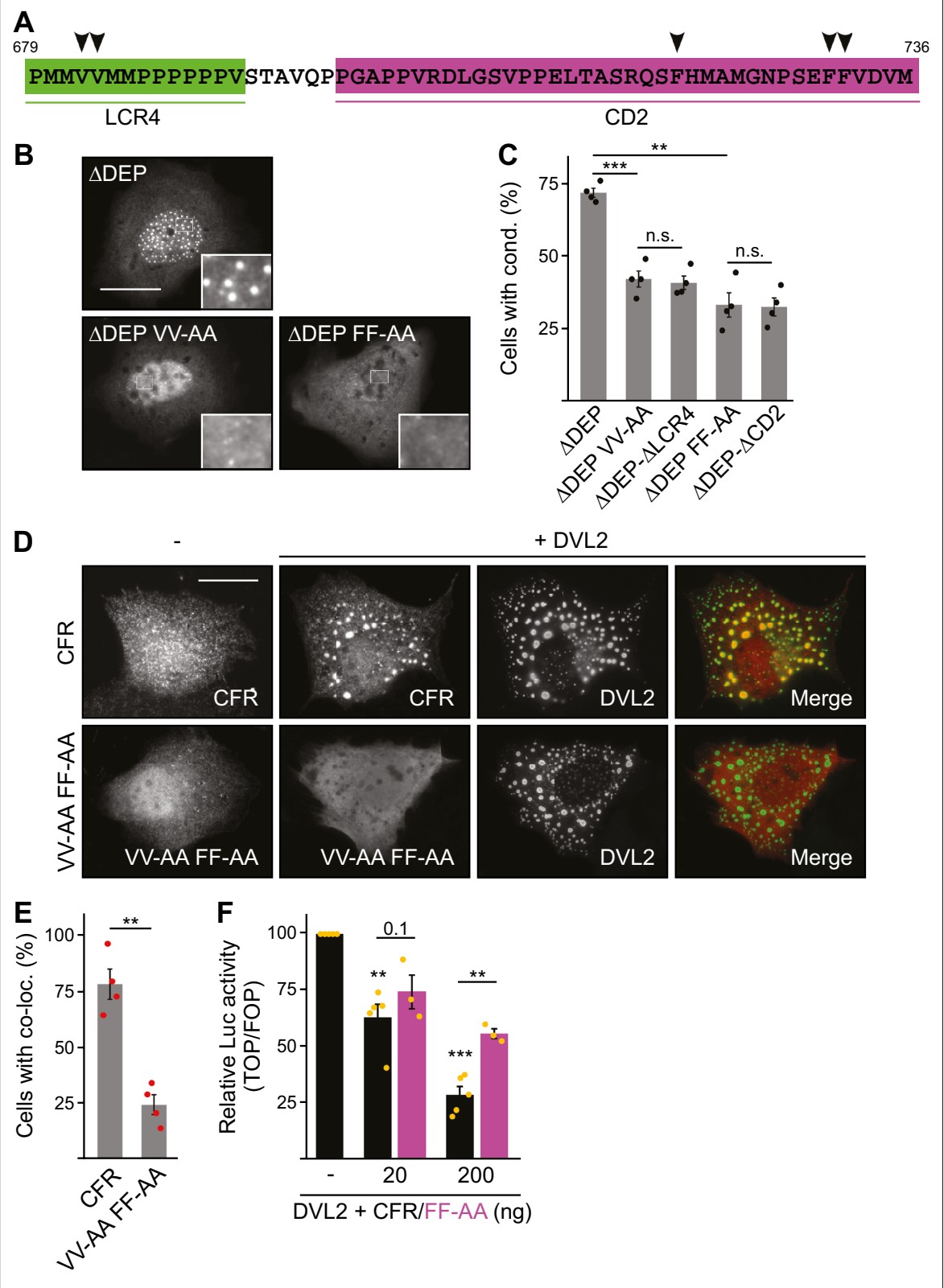

**Figure 5.** Low-complexity region (LCR4) and conserved domain (CD2) cooperatively mediate dishevelled 2 (DVL2)-DVL2 self-interaction. (**A**) Condensate forming region (CFR) amino acid sequence with highlighted LCR4 (green) and CD2 (magenta). Arrowheads point to residues potentially mediating protein-protein interaction. (**B**) Immunofluorescence of indicated HA-tagged proteins in transiently transfected U2OS cells. Scale bar: 20 μm. Insets are magnifications of the boxed areas. (**C**) Percentage of cells with condensates out of 1200 transfected cells from four independent experiments as in (**B**)

*Figure 5 continued on next page*

*Figure 5 continued*

(n=4). (**D**) Immunofluorescence of indicated proteins in U2OS cells, which were transfected with Flag-CFR or the Flag-CFR VV-AA FF-AA mutant either alone or in combination with DVL2. Scale bar: 20 µm. (**E**) Percentage of cells exhibiting co-localization of CFR or CFR VV-AA FF-AA with DVL2 out of 1200 transfected cells from four independent experiments as in (**D**) (n=4). (**F**) Relative luciferase activity reporting β-catenin-dependent transcription in HEK293T cells transfected with DVL2 alone or together with rising amounts of either CFR or the CFR FF-AA mutant (black bars [CFR] n=5, magenta bars [CFR FF-AA] n=3). (**C, E, F**) Results are mean ± SEM, **p<0.01, ***p<0.001 (Student's *t*-test).

The online version of this article includes the following source data and figure supplement(s) for figure 5:

**Source data 1.** Excel file providing the numerical source data to *Figure 5*.

**Figure supplement 1.** Specifying the residues mediating low-complexity region (LCR4) and conserved domain (CD2) function.

**Figure supplement 1—source data 1.** Excel file providing the numerical source data to *Figure 5—figure supplement 1*.

**Figure supplement 1—source data 2.** PDF files containing the original, labeled blots to *Figure 5—figure supplement 1*.

**Figure supplement 1—source data 3.** TIF files of the raw blots to *Figure 5—figure supplement 1*.

AXIN1-derived, sequence wise-nonrelated condensate-forming region (CFR$^{AX}$, see *Figure 6—figure supplement 1* for details) to DVL2 ΔDEP-ΔCFR (ΔDEP-ΔCFR+CFR$^{AX}$) restored condensate formation to the level of ΔDEP (*Figure 6D–F*), indicating that CFR$^{AX}$ compensates for loss of CFR$^{DVL2}$. More importantly, CFR$^{AX}$ fusion (ΔDEP-ΔCFR+CFR$^{AX}$) also rescued the decreased Wnt pathway activation of ΔDEP-ΔCFR compared to ΔDEP (*Figure 6G*), suggesting that it is indeed the CFR phase-separating activity that is crucial for signaling.

## DVL2 CFR contributes to Wnt pathway activation

In order to determine the impact of CFR within full-size DVL2, we mutated the four crucial residues identified in CFR in DLV2 (DVL2 VV-AA FF-AA). This shifted the DVL2 complexes in ultracentrifugation assays from large to smaller sizes, similar to deletion of CFR (*Figure 7A and B*), and reduced the formation of condensates by about 50% (*Figure 7C and D*). Importantly, DVL2 VV-AA FF-AA exhibited more than 50% reduced Wnt pathway activation compared to WT (*Figure 7E*), with a similar expression of both constructs (*Figure 5—figure supplement 1I*). The DVL2 variants were transiently expressed on top of endogenous DVL1/2/3 in this experiment. In addition, we used *DVL1/2/3* knockout cells, as they represent an elegant system to study Wnt pathway activation upon re-expression of DVL2 variants without any interference of endogenous WT DVL (*Paclíková et al., 2017*). In these cells, overexpression of DVL2 VV-AA FF-AA almost completely failed to activate the pathway and was as inactive as the DIX domain M2 mutant (*Schwarz-Romond et al., 2007a*), which can be considered as the gold standard for DVL2 inhibiting point mutations (*Figure 7F*). In addition, we used *DVL1/2/3* knockout cells with additional knockout of *RNF43* and *ZNRF3* (DVL tKO+), which allowed higher pathway activation upon DVL2 overexpression (*Figure 5—figure supplement 1J*), as the DVL2-activating receptors were no longer degraded through RNF43 and ZNRF3 (*Hao et al., 2012*; *Paclíková et al., 2021*). Also in these cells, DVL2 VV-AA FF-AA exhibited markedly impaired pathway activation as compared to WT (*Figure 5—figure supplement 1J*). Finally, we re-expressed DVL2 variants at close to endogenous levels in *DVL1/2/3* knockout cells to rescue Wnt-induced pathway activation, which was disrupted through DVL knockout (*Figure 7G*; *Figure 5—figure supplement 1K*). While re-expression of WT DVL2 resulted in a complete rescue as compared to WT cells, DVL2 VV-AA FF-AA was significantly impaired and as inactive as DVL2 M2 in this assay (*Figure 7G*). The VV-AA FF-AA mutation inhibited complexation, condensation, and Wnt pathway activation as efficiently as CFR deletion (*Figure 7A–F*), strongly indicating that it is specifically the interaction activity of CFR through the aggregon and the phenylalanine stickers that is required for signaling. A comparison between the VV-AA FF-AA mutation and the established M2 mutation showed on average about 65% and 80% reduction of Wnt pathway activation as compared to WT, respectively (*Figure 7E–G*; *Figure 5—figure supplement 1K*), suggesting that DVL2 CFR markedly contributes to Wnt pathway activation. Consistently, we observed strong significant correlations of CFR-mediated condensation and complexation with Wnt pathway activation for the DVL2 deletion constructs used in our study (*Figure 7H*).

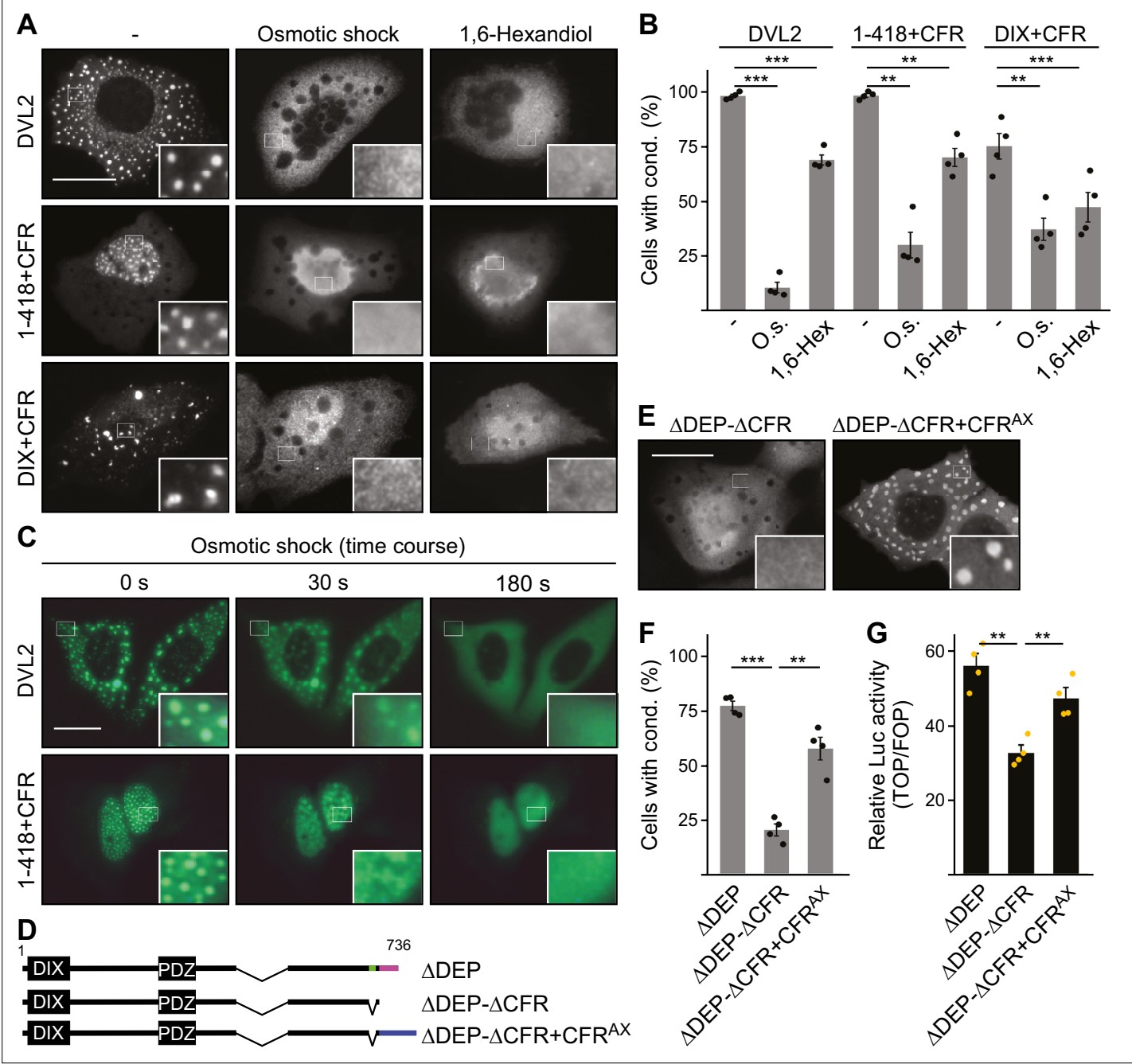

**Figure 6.** Condensate forming region (CFR)-induced condensates form via phase separation. (**A, E**) Immunofluorescence of indicated HA-tagged proteins in transiently transfected U2OS cells, which were untreated, exposed to osmotic shock for 3 min, or treated with 1 µM 1,6-hexanediol for 1 hr. Scale bars: 20 µm. Insets are magnifications of the boxed areas. (**B, F**) Percentage of cells with condensates out of 1200 transfected cells from four independent experiments as in (**A**) or (**E**) (n=4). (**C**) Fluorescence of indicated GFP-tagged proteins in transiently transfected, alive U2OS cells at different time points of osmotic shock treatment. Scale bar: 20 µm. Insets are magnifications of the boxed areas. (**D**) To scale schemes of dishevelled 2 (DVL2) constructs. Low-complexity region (LCR4), CD2, and CFR^AX are highlighted in green, magenta, and blue, respectively. (**G**) Relative luciferase activity reporting β-catenin-dependent transcription in HEK293T cells expressing the indicated constructs (n=4). (**B, F, G**) Results are mean ± SEM, **p<0.01, ***p<0.001 (Student's *t*-test).

The online version of this article includes the following source data and figure supplement(s) for figure 6:

**Source data 1.** Excel file providing the numerical source data to *Figure 6*.

**Figure supplement 1.** Identifying a phase-separating region in AXIN1.

**Figure supplement 1—source data 1.** Excel file providing the numerical source data to *Figure 6—figure supplement 1*.

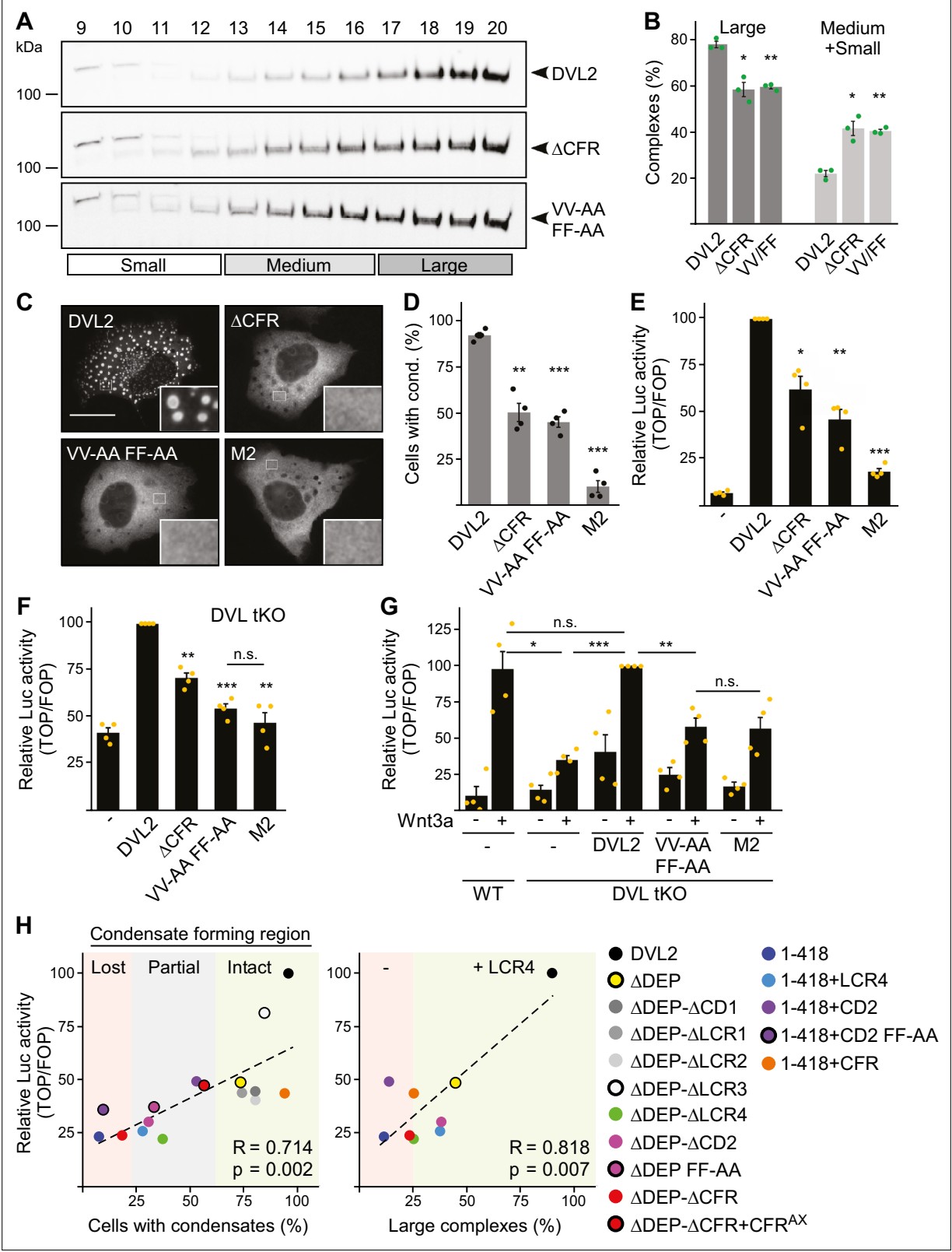

**Figure 7.** Dishevelled 2 (DVL2) condensate forming region (CFR) is crucial for Wnt signaling activity. (**A**) Western blotting for indicated proteins, which were transiently expressed in HEK293T cells, after fractionation of cell lysates via sucrose density ultracentrifugation (see **Figure 1E**). (**B**) Percentage of the protein that was engaged in large complexes or in medium/small complexes, as specified in (**A**) (n=3, refer to the legend in **Figure 1D** for more details). (**C**) Immunofluorescence of indicated HA-tagged DVL2 proteins in transiently transfected U2OS cells. Scale bar: 20 μm. Insets are magnifications

*Figure 7 continued on next page*

*Figure 7 continued*

of the boxed areas. (**D**) Percentage of cells with condensates out of 1200 transfected cells from four independent experiments as in (**C**) (n=4). (**E–G**) Relative luciferase activity reporting β-catenin-dependent transcription in U2OS cells (**E**) in T-REx cells with *DVL1/2/3* knockout (**F** DVL tKO) and in T-REx WT and DVL tKO cells (**G**), which were transiently transfected and treated with Wnt3a conditioned medium, as indicated (n=4). (**B, D–G**) Results are mean ± SEM, *p<0.05, **p<0.01, ***p<0.001 (Student's *t*-test). (**H**) Plotting of Wnt pathway activation (y-axis) against either condensation (x-axis; left side) or complexation (x-axis; right side) for indicated DVL2 wild-type (WT) and mutant proteins. Correlation strength and significance are indicated by the Pearson's correlation coefficient R and the p-value, respectively. Note that condensation correlates with whether CFR is intact (low-complexity region, LCR4 and conserved domain, CD2 intact, green), partially intact (either LCR4 or CD2 intact, gray), or lost (neither LCR4 nor CD2 intact, red), and that the presence of LCR4 determines complexation. The plots summarize data that were shown before within this study.

The online version of this article includes the following source data for figure 7:

**Source data 1.** Excel file providing the numerical source data to *Figure 7*.

**Source data 2.** PDF file containing the original, labelled blots to *Figure 7*.

**Source data 3.** TIF files of the raw blots to *Figure 7*.

## Discussion

Here, we provided strong biochemical evidence that endogenous DVL2 forms oligomeric complexes (*Figure 1A and B*), supporting the idea that DVL2 assemblies exist at physiologic protein levels. Although we investigated several scaffold proteins with various endogenous interactors in this assay, we only observed such complexes for DVL2 and AXIN2 and they were specifically associated with aggregating protein sequences (*Bernkopf et al., 2019*; *Miete et al., 2022*), indicating that most types of protein-protein interactions are not preserved in this assay. Furthermore, overexpressed DVL2 did not exhibit reduced complex sizes, as one would have expected, if limited endogenous interactors had been part of the complexes. Therefore, we think that the detected complexes most likely reflect homotypic DVL2 assemblies, which would then be about eight molecules in size (*Figure 1A and B*). The size of the complexes was reminiscent of previously described endogenous DVL2 oligomers identified via TIRF imaging (*Kan et al., 2020*). Through deletion analysis, we discovered a 14 aa long LCR4 in the DVL2 C-terminus mediating complex formation (*Figure 4A–D*) and condensate formation (*Figure 3—figure supplement 1B*). An adjacent 38 aa long, evolutionary CD2 was required for the formation of DVL2 condensates (*Figure 3—figure supplement 1B*), and we conceptually combined LCR4 and CD2 as CFR. Molecularly, an aggregon in LCR4 and phenylalanine residues in CD2 mediated DVL2 assembly (*Figure 7A–D*; *Figure 5—figure supplement 1A–G*). The latter may promote protein interaction via sticking of their aromatic rings, as previously described for phase-separating proteins (*Martin et al., 2020*). Co-localization of the isolated CFR with DVL2 indicated that CFR may mediate DVL2-DVL2 interaction, and this was prevented by specific point mutations targeting the aggregon in LCR4 and the phenylalanine stickers in CD2 (*Figure 5D and E*). Treatments that challenge phase separation diffused CFR-induced condensates (*Figure 6A–C*), in line with a recent report showing that DVL2 condensates form via phase separation (*Kang et al., 2022*). Importantly, point mutations that inhibit CFR self-interaction markedly attenuated Wnt pathway activation by DVL2 (*Figure 7E–G*; *Figure 5—figure supplement 1K*). Especially in *DVL1/2/3* triple knockout cells, the DVL2 CFR point mutant VV-AA FF-AA was as signaling deficient as the DIX domain M2 mutant (*Figure 7F and G*), which is frequently used for DVL2 inactivation (*Schwarz-Romond et al., 2007a*). In these cells, mere DVL2 VV-AA FF-AA overexpression almost completely failed to activate the pathway (*Figure 7F*), and its re-expression at close to endogenous levels only poorly rescued activation of the pathway through Wnt ligands, which was disrupted by the DVL knockout (*Figure 7G*; *Figure 5—figure supplement 1K*). We thus identified a novel DVL2 region that promotes complex formation, condensate formation, and Wnt pathway activation.

Our study provides deeper insights into the assembly of multimeric DVL2 condensates.

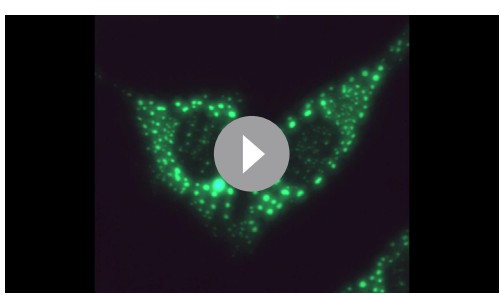

**Video 1.** Osmotic shock dissolves dishevelled 2 (DVL2) condensates. Fluorescence of GFP-DVL2 in transiently transfected, alive U2OS cells, which were imaged every 15 s over 3 min of osmotic shock treatment.

https://elifesciences.org/articles/96841/figures#video1

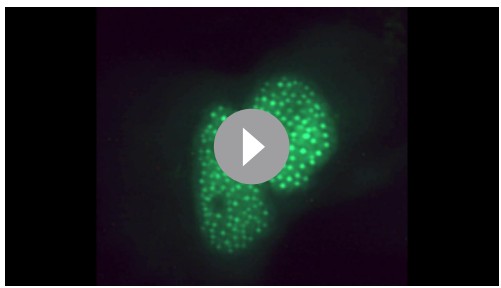

**Video 2.** Osmotic shock dissolves condensate forming region (CFR)-induced condensates. Fluorescence of GFP-1–418+CFR in transiently transfected, alive U2OS cells, which were imaged every 15 s over 3 min of osmotic shock treatment.

https://elifesciences.org/articles/96841/figures#video2

The complexes detected by ultracentrifugation showed sizes of about eight DVL2 molecules, which were irrespective of expression levels as they were similar between endogenous and over-expressed DVL2 (*Figure 1A–C*). Therefore, the complexes cannot be identical with the known microscopically visible condensates, containing thousands of molecules. These oligomeric complexes thus existed in parallel to the condensates or constituted a condensate substructure. Analysis of mutant DVL2 proteins revealed that DVL2 contains domains that promote both complexes and condensates, such as LCR4 and DEP (*Figure 2B–E*; *Figure 4A-D*, *Figure 3—figure supplement 1B*), and domains that promote only condensates but not complexes, such as CD2 and DIX (*Figure 1C*; *Figure 4A–D*; *Figure 7C*, *Figure 3—figure supplement 1B*). However, all DVL2 mutations that reduced complexes also affected condensates. Therefore, we suggest that the oligomeric complexes are required for and possibly represent substructures of the multimeric condensates (*Figure 8*). As the stability of complexes at low protein concentrations, e.g., in cellular extracts, indicated a rather high affinity of the underlying interaction sites (*Figure 1A*), we hypothesize that complex formation via LCR4 and DEP precedes further assembly of these substructures into condensates via CD2 and DIX (*Figure 8*). Our proposed two-step model of DVL2 condensate formation is in line with and supportive of the emerging stickers model for the formation of biomolecular condensates (*Choi et al., 2020*). According to this model, oligomerization via one interaction site can drive subsequent condensate formation by increasing the valence of another interaction site (=sticker) of the oligomer compared to a monomer (*Choi et al., 2020*). The proposed two-step model would increase the options for regulating the polymerization of DVL2 by modulating the oligomerization step or the condensate formation step (*Figure 8*). DVL2 polymerization is crucial for Wnt signal transduction. However, the low DVL2 concentration and the low DIX-DIX affinity strongly disfavor polymerization, and clustering of DVL2 at activated Wnt-receptor complexes is suggested to overcome this limitation (*Bienz, 2014*). Pre-oligomerization of DVL2 via high-affinity interactions as identified in the ultracentrifugation assay might facilitate Wnt-induced polymerization.

Furthermore, the discovered C-terminal self-interaction site CFR may contribute to the regulation of DVL2 condensates through conformational changes. Binding of the very C-terminus of DVL2 to its PDZ domain results in a closed protein conformation (*Lee et al., 2015*). It has been suggested that this closed conformation limits the accessibility of the N-terminal intrinsically disordered region that promotes condensate formation, thereby suppressing DVL2 condensates (*Kang et al., 2022*;

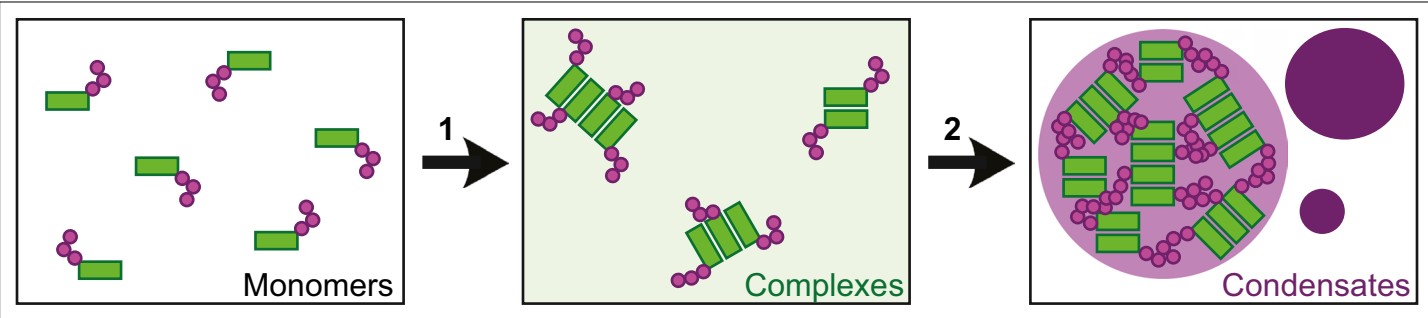

**Figure 8.** Two-step model of DVL2 condensate formation. Schematic illustration of DVL2 domains mediating the formation of oligomeric complexes, such as LCR4 and the DEP domain (green rectangles) and of sticker domains mediating condensate formation, such as CD2 and the DIX domain (magenta circles). Oligomerization into complexes (1) (light green background) increases the valence of stickers in the complexes as compared to the monomers, which allows to overcome the low affinity of the isolated stickers and drives subsequent formation of condensates (2) (purple) by the multivalent stickers, according to the emerging stickers-model (*Choi et al., 2020*).

*Vamadevan et al., 2022*). It is very intriguing to speculate that condensate formation will be additionally suppressed in the closed conformation by limiting the accessibility of CFR because only one amino acid separates the identified crucial phenylalanine residues of CFR from the PDZ binding motif in the DVL2 C-terminus. Notably, Wnt ligands induce the open DVL conformation (*Harnoš et al., 2019*), which will increase the CFR accessibility and, according to our model, would allow LCR4 and CD2 to contribute to pre-oligomerization and subsequent condensate formation of DVL2, respectively.

Several studies suggest functional differences between the DVL paralogs (*Gentzel et al., 2015*; *Lee et al., 2008*; *Paclíková et al., 2021*), while the underlying molecular differences remain unclear. Notably, complex formation as revealed by ultracentrifugation was only observed for DVL2 and not for DVL1 or DVL3, revealing a marked difference between the paralogs (*Figure 1C*). In addition, CFR did not co-localize with DVL1 or DVL3 in contrast to DVL2, indicating that CFR-mediated DVL2-DVL2 interaction, which most likely drove DVL2 complexation, is absent from DVL1 and DVL3 (*Figure 5D*; *Figure 5—figure supplement 1H*). Consistently, the crucial CFR aggregon located in LCR4 was not conserved in DVL1 or DVL3 (*Figure 1—figure supplement 1D*). Moreover, replacing the complementary part of DVL1 with the DVL2 CFR promoted complex formation (*Figure 4E, F*). Although DVL1 and DVL3 lack the discovered CFR, all three DVL paralogs are able to form condensates (*Figure 2D*; *Figure 5—figure supplement 1H*) and to activate Wnt signaling to some extent (*Lee et al., 2008*; *Paclíková et al., 2021*), which can be potentially explained by the other interaction sites (DIX, DEP, intrinsically disordered region). However, quantitative studies suggest functional differences between the paralogs and that they have to cooperate at a certain molar ratio for optimal Wnt pathway activation, with DVL2 being the most abundant (*Lee et al., 2008*; *Paclíková et al., 2021*). In this context, the DVL2 condensates with their underlying stable complexes may function as a kind of super scaffold for the integration of DVL1 and DVL3. Our findings may help to understand the functional differences between the paralogs in the future.

## Materials and methods

**Key resources table**

| Reagent type (species) or resource | Designation | Source or reference | Identifiers | Additional information |
|---|---|---|---|---|
| Gene (*Homo sapiens*) | *DVL2* | GenBank | 1856 | |
| Cell line (*Homo sapiens*) | HEK293T | ATCC | CRL-3216 | |
| Cell line (*Homo sapiens*) | HeLa | ATCC | CCL-2 | |
| Cell line (*Homo sapiens*) | U2OS | ATCC | HTB-96 | |
| Cell line (*Homo-sapiens*) | T-REx | *Paclíková et al., 2017* | | |
| Cell line (*Homo-sapiens*) | DVL tKO | *Paclíková et al., 2017* | | T-REx cells with *DVL1/2/3* triple knockout |
| Cell line (*Homo-sapiens*) | DVL tKO+ | *Paclíková et al., 2021* | | T-REx cells with *DVL1/2/3*, *RNF43* and *ZNRF3* penta knockout |
| Antibody | anti-DVL2 (Rabbit polyclonal) | CellSignaling | Cat# 3216 S; | WB (1:1000) |
| Antibody | anti-DVL2 (Rabbit polyclonal) | CellSignaling | Cat# 3224 S | WB (1:1000) |
| Antibody | anti-AXIN1 (Rabbit polyclonal) | CellSignaling | Cat#: 2087 S | WB (1:1000) |
| Transfected construct (human) | siRNA to DVL2 | Dharmacon/Thermo Fisher Scientific; *Soh and Trejo, 2011* | | GGAAGAAAUUUCAGAUGAC |
| Recombinant DNA reagent | HA-DVL2 (plasmid) | *Bernkopf et al., 2015* | | |

*Continued on next page*

*Continued*

| Reagent type (species) or resource | Designation | Source or reference | Identifiers | Additional information |
|---|---|---|---|---|
| Recombinant DNA reagent | Expression plasmids for deleted and point mutated HA-tagged DVL2 | This paper | | More than 25 plasmids have been newly generated (please see Molecular Biology for details) |
| Chemical compound, drug | Sucrose | Fluka Analytical | 84097 | |
| Chemical compound, drug | Thyroglobulin (669 kDa) | Sigma Aldrich | T9145 | Size marker |
| Chemical compound, drug | Albumin (66 kDa) | Sigma Aldrich | A8531 | Size marker |
| Chemical compound, drug | 1,6-Hexanediol | Sigma Aldrich | 240117 | |
| Software, algorithm | TANGO algorithm | http://tango.crg.es/; *Fernandez-Escamilla et al., 2004* | | Prediction of aggregation sites |
| Software, algorithm | Spot Detector tool, Icy bio-imaging software | Institut Pasteur, version 2.2.1.0; *Olivo-Marin, 2002* | | Quantify the numbers of condensates per cell |
| Software, algorithm | catGRANULE algorithm | http://s.tartaglialab.com/update_submission/907235/f570b07a95; *Bolognesi et al., 2016* | | Prediction of liquid-liquid phase separation propensity |
| Other | Wnt3a medium | *Willert et al., 2003* | | Wnt3a medium was prepared following the published protocol |

## Cell culture, transfections, and treatments

HEK293T, HeLa, and U2OS cells were grown in low glucose DMEM supplemented with 10% fetal calf serum and antibiotics at 37 °C in a 10% $CO_2$ atmosphere, and passaged according to ATCC recommendations, where the cell lines were originally obtained from. Cell lines were authenticated based on cell morphology and size, and cells were tested negative for mycoplasma contamination. T-REx *DVL1/2/3* triple knockout cells (DVL tKO) and T-REx *DVL1/2/3, RNF43,* and *ZNRF3* penta knockout cells (DVL tKO+) were generated in the Bryja lab and have been previously described (*Paclíková et al., 2017*; *Paclíková et al., 2021*). They were grown in high glucose DMEM supplemented with GlutaMAX, 10% fetal calf serum, and antibiotics at 37 °C in a 10% $CO_2$ atmosphere. Transfections of plasmids and siRNA were performed with polyethylenimine and Oligofectamine (Invitrogen) according to the manufacturer's recommendations, respectively. Wnt3a medium was prepared as originally described (*Willert et al., 2003*). For the osmotic shock treatment, cell culture medium was diluted 1:1 with sterile water to reduce the osmolarity by 50%. 1,6-Hexanediol (240117) was obtained from Sigma-Aldrich.

## Molecular biology

The expression vectors for HA-DVL1, HA-DVL2, HA-DVL3, and HA-DVL2 M2 have been described previously (*Bernkopf et al., 2015*). The expression vectors for HA-ΔDEP, HA-ΔDEP-ΔCD1, HA-ΔDEP-ΔCD2, HA-ΔDEP-ΔLCR1, HA-ΔDEP-ΔLCR2, HA-ΔDEP-ΔLCR3, HA-ΔDEP-ΔLCR4, HA-ΔDEP-ΔCFR, HA-ΔDEP-ΔCFR+CFR^AX, HA-1–418, HA-1–418+CD2, HA-1–418+LCR4, HA-1–418+CFR, HA-DVL1-CFR^DVL2, HA-DIX, HA-DIX+CFR, HA-ΔCFR, Flag-DAX, Flag-CFR+DAX, Flag-1xCFR^AX-DAX, Flag-2xCFR^AX-DAX, Flag-3xCFR^AX-DAX, Flag-CFR were cloned via standard molecular biology methods. HA-1–418+CFR M2, HA-DIX+CFR M2, Flag-CFR+DAX M3, Flag-3xCFR^AX-DAX M3, HA-ΔDEP VV-AA, HA-ΔDEP FF-AA, HA-1–418+LCR4 VV-AA, HA-1–418+CD2 FF-AA, Flag-CFR FF-AA, Flag-CFR VV-AA FF-AA, HA-DVL2 VV-AA FF-AA were generated using site-directed mutagenesis. All generated expression vectors were verified by sequencing. The newly created expression vectors are available from the corresponding author on request.

## Antibodies and siRNA

We used the following antibodies in this study: Primary antibodies: rb α DVL2 [WB: 1:1000], 3216 S; rb α DVL2 [WB: 1:1000], 3224 S; rb α Axin1 [WB: 1:1000], 2087 S CellSignaling / rat α HA [WB:

1:1000], 11867423001 Roche / rat αα-tubulin [WB: 1:1000], MCA77G Serotec / m α Flag [IF: 1:800], F3165; rb α Flag [IF: 1:300], F7425; rb α HA [IF: 1:200], H6908 Sigma-Aldrich. Secondary antibodies: goat α mouse/rabbit-Cy3 [1:300], goat α rabbit-Cy2 [1:200], goat α mouse/rabbit/rat-HRP [1:2000] (Jackson ImmunoResearch). The siRNA targeting human DVL2 (5'-GGAAGAAAUUUCAGAUGAC-3') was published (*Soh and Trejo, 2011*).

## Sucrose gradient ultracentrifugation

Cells were lysed about 24 hr after transfection, when required, or 48 h after seeding in a Triton X-100-based buffer (150 mM NaCl, 20 mM Tris-HCl pH 7.5, 5 mM EDTA, 1% Triton X-100, Roche protease inhibitor cocktail). A linear sucrose density gradient was prepared in 13×51 mm centrifuge tubes (Beckman Coulter) by overlaying 2 ml of a 50% (v/w) sucrose solution with 2 ml of a 12.5% (v/w) sucrose solution followed by horizontal incubation of the tube for 3 h at RT (*Stone, 1974*), bevor loading a 200 µl cell lysate sample on top. After centrifugation in a Beckman Coulter Optima MAX Ultracentrifuge (217100 g, 25 °C, 18 hr), 20 fractions à 200 µl were collected from top to bottom and analyzed by Western blotting, as indicated. The commercially available size markers thyroglobulin (669 kDa, T9145) and albumin (66 kDa, A8531) were obtained from Sigma Aldrich. In the case of thyroglobulin and albumin, fractions were analyzed by silver staining of the proteins in polyacrylamide gels.

## Western blot

Proteins in cell lysates or in fractions of sucrose gradients were denatured, separated by gel electrophoresis in polyacrylamide gels under denaturing conditions (SDS-PAGE), and transferred onto a nitrocellulose membrane (VWR). The proteins were detected using suitable combinations of primary and HRP-conjugated secondary antibodies (see above) via light emission upon HRP-catalyzed oxidation of luminol in a LAS-3000 with Image Reader software (FUJIFILM). Intensities of protein bands were quantified with AIDA 2D densitometry.

## Immunofluorescence

Cells were fixed in a 3% paraformaldehyde solution, permeabilized with 0.5% Triton X-100, and blocked with cell culture medium to reduce unspecific antibody binding, before proteins of interest were stained with suitable combinations of primary and fluorochrome-conjugated secondary antibodies (see above). Analysis and image acquisition was performed at an Axioplan II microscope system (Carl Zeiss) using a Plan-NEOFLUAR 100 x/1.30 NA oil objective and a SPOT RT Monochrome camera (Diagnostic Instruments). Cells were categorized in a blinded fashion as 'cell with condensates' when exhibiting more than three distinct sphere-like structures, to reduce the number of false positives. The Spot Detector tool of the Icy open source bio-imaging software (Institut Pasteur, version 2.2.1.0) was used to objectively quantify the numbers of condensates per cell (*Olivo-Marin, 2002*).

## Live-cell imaging

For live-cell imaging of the osmotic shock treatment, the culture medium of cells expressing indicated GFP-tagged proteins was replaced with a 50% hypoosmolar phosphate buffered saline solution. Images were acquired at constant exposure times every 15 s over the next 3 min at an Axiovert25 microscope system (Carl Zeiss) using a LD A-Plan 40 x/0.50 Ph2 objective and a SPOT Insight QE camera with the SPOT Basic software (Diagnostic Instruments, version 4.0.1). Videos were rendered using Photoshop 19.1.5 (Adobe).

## Luciferase reporter assay

Cells were transfected with a luciferase reporter plasmid either with a β-catenin-dependent promoter (TOP, Tcf optimal) or with a β-catenin-independent control promoter (FOP, far from optimal), a constantly active β-galactosidase expression plasmid and expression plasmids for the proteins of interest, as indicated in the figures. After lysis (25 mM Tris-HCl pH 8, 2 mM EDTA, 5% glycerol, 1% Triton X-100, 20 mM DTT), the luciferase activity was measured via light emission upon luciferin decarboxylation in a Centro LB 960 Microplate Luminometer (Berthold technologies) and the β-galactosidase activity was assessed as a release of yellow ortho-nitrophenol upon ortho-nitrophenyl-β-galactoside hydrolysis using a Spectra MAX 190 (Molecular Devices). The luciferase activities were first normalized to the respective β-galactosidase activities to correct for minor variations in transfection efficiency, before

the TOP values were normalized to the respective FOP values to correct for unspecific β-catenin-independent changes. TOP/FOP reporter assays were performed in technical duplicates.

## Statistical analysis

Data sets were probed for statistical significance using two-tailed Student's $t$-tests in a non-paired (*Figure 3D*; *Figure 3—figure supplement 1E–G*) or paired (all others) fashion, depending on the experimental setup. Statistical significance is indicated by asterisks in the figures (*p<0.05, **p<0.01, ***p<0.001), when required, and n values of biological replicates are stated in the figure legends for all experiments. We assumed a normal distribution of the data based on the nature of the assays and graphical assessment, which, however, was not formally tested owing to the small sample sizes. p-values for the correlation analyses in *Figure 7H* were calculated by the test statistic $t=R*$square root$((n\text{-}2)/(1\ R^2))$, with n-2 degrees of freedom.

## Acknowledgements

The authors thank Gabriele Daum for excellent technical assistance. This study was funded by grants from the Deutsche Forschungsgemeinschaft to JB (BE 1550/12–1) and DBB (BE 7055/2–1), from the Wilhelm Sander-Stiftung to DBB (2018.017.2), and from the Friedrich-Alexander University Erlangen-Nürnberg Interdisciplinary Center for Clinical Research to JB and DBB (D30). VB was supported by the Czech Science Foundation grant no. GA22-25365S.

## Additional information

### Funding

| Funder | Grant reference number | Author |
| --- | --- | --- |
| Deutsche Forschungsgemeinschaft | BE 1550/12-1 | Jürgen Behrens |
| Deutsche Forschungsgemeinschaft | BE 7055/2-1 | Dominic B Bernkopf |
| Wilhelm Sander-Stiftung | 2018.017.2 | Dominic B Bernkopf |
| Interdisciplinary Center for Clinical Research, Erlangen | D30 | Dominic B Bernkopf |
| Czech Science Foundation | GA22-25365S | Vítězslav Bryja |

The funders had no role in study design, data collection and interpretation, or the decision to submit the work for publication.

### Author contributions

Senem Ntourmas, Formal analysis, Methodology; Martin Sachs, Martina Brückner, Methodology; Petra Paclíková, Resources, Formal analysis, Writing – review and editing; Vítězslav Bryja, Resources, Formal analysis, Funding acquisition, Writing – review and editing; Jürgen Behrens, Conceptualization, Funding acquisition, Writing – review and editing; Dominic B Bernkopf, Conceptualization, Formal analysis, Funding acquisition, Methodology, Writing – original draft, Writing – review and editing

### Author ORCIDs

Vítězslav Bryja ⬚ https://orcid.org/0000-0002-9136-5085
Dominic B Bernkopf ⬚ https://orcid.org/0000-0003-4870-5248

Reviewer #2 (Public review): https://doi.org/10.7554/eLife.96841.3.sa1
Author response https://doi.org/10.7554/eLife.96841.3.sa2

## Additional files

### Supplementary files
- MDAR checklist

### Data availability
All data generated or analyzed during this study are included in the manuscript and supporting files; numerical source data files have been provided for Figures 1–7 and for Figure 3—figure supplement 1, Figure 3—figure supplement 2, Figure 5—figure supplement 1 and Figure 6—figure supplement 1, and source data files for blots and gels have been provided for Figures 1, 2, 4 and 7 and for Figure 1—figure supplement 1 and Figure 5—figure supplement 1.

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
