## [Editor Report · eLife Assessment]

This **valuable** study contributes to the understanding of phase separation in Dishevelled (DVL) proteins, by investigating the endogenous complexes of DVL2 using ultracentrifugation and contrasting them with DVL1 and DVL3 behaviour and the functional validation of the DVL2 intrinsically disordered regions mediating the protein condensate. The study includes a **solid** characterisation of several overexpression constructs, including in KO cells. However, investigations of the roles of the described DVL2 regions at the endogenous level remain to be carried out.

---

## [Referee Report · Reviewer #2 (Public review)]

Summary:

The authors aimed to identify which regions of DVL2 contribute to its endogenous/basal clustering, as well as the relevance of such domains to condensate/phase separation and WNT activation.

Strengths:

A strength of the study is the focus on endogenous DVL2 to set up the research questions, as well as the incorporation of various techniques to tackle it. I found also quite interesting that DVL2-CFR addition to DVL1 increased its MW in density gradients.

Weaknesses:

The authors have addressed important drawbacks regarding the overexpression experiments, most notably by including DVL tKO cells in collaboration with Vita. I think that this part has clearly improved.

Unfortunately, I still stand with my key concern: at this stage in the field, with many papers on DVL over expression, there is a clear need to address how endogenous DVL behaves. While the attempts to o/e low levels of DVL mutants in tKO cells are useful for validation experiments, the manuscript does not -in my opinion - address the requirements of DVL2 condensation for WNT signalling. Of note, several of the described effects are partial, including in tKO cells, often indicating that the targeted domains contribute, but are not required, for these processes. I understand that generating endogenous tagged lines or targeting specific endogenous domains is not trivial. But, as indicated in both initial reviews, I think that monitoring endogenous proteins is required to fully address the proposed research question.

In my opinion, the current manuscript (A) shows that endogenous DVL2 forms large complexes in a higher proportion as DVL1/3, (B) identifies and describes a couple of motifs that contribute to clustering and signalling in overexpressed DVL, including in tKO cells* (C) shows that one of those motifs (CFR) rewires o/e DVL1 into behaving similarly as DVL2.

*On a minor note, I am not sure how DVL tKO cells partially react to Wnt3a in Figure 7G

---

## [Author Response]

The following is the authors’ response to the original reviews.

**Public Reviews:**

**Reviewer #1 (Public Review):**
Summary:The study "Endogenous oligomer formation underlies DVL2 condensates and promotes Wnt/βcatenin signaling" by Senem Ntourmas et al. contributes to the understanding of phase separation in Dishevelled (DVL) proteins, specifically focusing on DVL2. It builds upon existing research by investigating the endogenous complexes of DVL2 using ultracentrifugation and contrasting them with DVL1 and DVL3 behavior. The study identifies a DVL2-specific region involved in condensate formation and introduces the "two-step" concept of DVL2 condensate formation, enriching the field's knowledge.Strengths:A notable strength of this study is the validation of endogenous DVL2 complexes, providing insights into its behavior compared to DVL1 and DVL3. The functional validation of the DVL C-terminus here termed conserved domain 2 (CD2) and the identification of DVL2-specific regions (here termed LCR4) involved in condensate formation are significant contributions that complement the current knowledge on the importance of DVL DIX domain, DEP domain and intrinsically disordered regions between DIX and PDZ domains. Additionally, the introduction of the concept where oligomerization (step 1) precedes condensate formation (step 2) is an interesting hypothesis, which can be further experimentally challenged in the future.

We thank the reviewer for her/his interest in our work and for acknowledging our significant contributions to the understanding of DVL2 phase separation.

Weaknesses:However, the applicability of the findings to full-length DVL2 protein, hence the physiological relevance, is limited. This is mostly due to the fact that the authors almost completely depend on the set of DVL2 mutants, which lack the (i) DEP domain and (ii) nuclear export signal (NES). These variants fail to establish DEP domain-mediated interactions, including those with FZD receptors. Of note, the DEP domain itself represents a dimerization/tetramerization interface, which could affect the protein condensate formation of these mutants. Possibly even more importantly, the used mutants localize into the nucleus, which has different biochemical & biophysical properties than a cytoplasm, where DVL typically reside, which in turn affects the condensate formation. On top, in the nucleus, most of the DVL binding partners, including relevant kinases, which were reported to affect protein condensate formation, are missing.

The most convincing way to address this valid concern and to support a physiological relevant role of our findings is to extend our experiments with full-length DVL2, which we did alongside the suggestion in point two (please see below). In addition, we address the specific issues as follows:

We completely agree that interaction through the DEP domain contributes to condensate formation, which was thoroughly demonstrated in great studies by Melissa Gammons and Mariann Bienz, and complex formation (Fig. 2B, C). We deleted this domain on purpose for our mapping experiments, since we obtained more consistent results without any additional contribution of the DEP domain. Once we mapped CFR and identified crucial amino acids within CFR (VV, FF), we demonstrated that CFR-mediated interaction contributes to complex formation, condensate formation and pathway activation in the context of full-length DVL2 (Fig. 7A-G).

We also agree that the nuclear localization may affect condensate formation because of the reasons mentioned by the reviewer or others, such as differences in DVL2 protein concentration. However, later proof-of-concept experiments in full-length DVL2 confirmed that CFR and its identified crucial amino acids (VV, FF), which were mapped in this rather artificial nuclear context, contribute to the typical cytosolic condensate formation of DVL2 (Fig. 7C, D). Moreover, we also observed cells with cytosolic condensates for the NES-lacking DVL2 constructs, although to a lower extent as compared to cells with nuclear condensates. A new analysis of NES-lacking key constructs focusing exclusively on cells with cytosolic condensates revealed similar differences between the DVL2 mutants as were observed before when investigating cells with nuclear (and cytosolic) condensates (new Fig. S3E, F), suggesting that the detected differences are not due to nuclear localization but reflect the overall condensation capacity.

In addition, our condensate-challenging experiments (osmotic shock, 1,6-hexandiol) suggested that cytosolic condensates of full-length DVL2 and nuclear CFR-mediated condensates of deletion proteins lacking the DEP domain behave quite similar (Fig. 6A-C).

Second, the use of an overexpression system, while suitable for comparing DVL2 protein condensate features, falls short in functional assays. The study could benefit from employing established "rescue systems" using DVL1/2/3 knockout cells and re-expression of DVL variants for more robust functional assessments.

We used the suggested established rescue system of DVL1/2/3 knockout cells (T-REx *DVL1/2/3* triple knockout cells and T-REx *DVL1/2/3 RNF43 ZNRF3* penta knockout cells, which are even more sensitive towards DVL re-expression as they lack RNF43/ZNRF3-mediated degradation of DVL activating receptors; both cell lines from the Bryja lab). Upon overexpression, our key mutants DVL2 VV-AA FF-AA and ∆CFR showed markedly reduced pathway activation compared to WT DVL2 (new Figs. 7F and S5J), as we observed before. Especially in the *DVL1/2/3* triple knockout cells, DVL2 VV-AA FF-AA hardly activated the pathway and was as inactive as the established M2 mutant (new Fig. 7F). Most importantly, while re-expression of WT DVL2 at close to endogenous expression levels fully rescued Wnt3a-induced pathway activation in *DVL1/2/3* knockout cells, DVL2 VV-AA FF-AA revealed significantly reduced rescue capacity and was almost as inactive as DVL2 M2 (new Figs. 7G and S5K).

Furthermore, the discussion and introduction overlook some essential aspects of DVL biology. One such example is the importance of the open/close conformation of DVL and its effects on DVL phase separation and activity. In the context of this study, it is important to say that this conformational plasticity is mediated by DVL C-terminus (CD2 in this study). The second example is the reported roles of DVL1 and DVL3, which can both mediate the Wnt3a signal. How this can be interpreted when DVL1 and DVL3 lack LCR4 and still form condensates?

We included the open/close conformation of DVL in our manuscript (introduction p. 3 and new discussion paragraph p. 10) and discussed it in the context of our findings. It is intriguing to speculate that Wnt-induced opening of DVL2 increases the accessibility of LCR4 and CD2, thereby triggering pre-oligomerization and subsequent phase separation of DVL2 (see discussion).

We extended the last paragraph of the discussion to interpret the roles of DVL1 and DVL3 lacking LCR4 (see p. 10). In short, the general ability of DVL1 and DVL3 to form condensates and to activate the Wnt pathway can be potentially explained through the other interaction sites (DIX, DEP, intrinsically disordered region). However, previous studies suggest that the DVL paralogs exhibit (quantitative) differences in Wnt pathway activation and that all three paralogs have to interact at a certain ratio for optimal pathway activation. In this context, a physiologic role for DVL2 LCR4 may be to promote the formation of these DVL1/2/3 assemblies and/or to enhance the stability of these assemblies.

In order to increase the physiological relevance of the study, I would recommend analyzing several key mutants in the context of the full-length DVL2 protein using the rescue/complementation system. Further, a more thorough discussion and connections with the existing literature on DVL protein condensates/puncta/LLPS can improve the impact of the study.

We thank the reviewer for her/his suggestions to improve our study, which we addressed as detailed above.

**Reviewer #2 (Public Review):**
Summary:The authors aimed to identify which regions of DVL2 contribute to its endogenous/basal clustering, as well as the relevance of such domains to condensate/phase separation and WNT activation.Strengths:A strength of the study is the focus on endogenous DVL2 to set up the research questions, as well as the incorporation of various techniques to tackle it. I found also quite interesting that DVL2-CFR addition to DVL1 increased its MW in density gradients.

We thank the reviewer for her/his interest in our work and the constructive suggestions to improve our study.

Weaknesses:I think that several of the approaches of the manuscript are subpar to achieve the goals and/or support several of the conclusions. For example:(1) Although endogenous DVL2 indeed seems to form complexes (Figure 1A), neither the number of proteins involved nor whether those are homo-complexes can be determined with a density gradient. Super-resolution imaging or structural analyses are needed to support these claims.

We agree that it will be very interesting to study the nature of the detected endogenous complexes in detail and we will consider this for any follow-up study, as structural analyses were out of scope for the revision of the presented manuscript. To address the issue, we mentioned that the calculation of about eight DVL2 molecules per complex is based on the assumption of homotypic complexes (results p. 4) and we discussed, why we think that homotypic complexes are the most likely assumption based on the currently available (limited) data (discussion p. 8).

(2) Follow-up analyses of the relevance of the DVL2 domains solely rely on overexpressed proteins. However, there were previous questions arising from o/e studies that prompted the focus on endogenous, physiologically relevant DVL interactions, clustering, and condensate formation.Although the title, conclusions, and relevance all point to the importance of this study for understanding endogenous complexes, only Figures 1A and B deal with endogenous DVL2.

We think that the biochemical detection of endogenous DVL2 complexes itself represents a valuable contribution to the understanding of endogenous DVL clustering, especially (i) since it is still lively discussed in the field whether and to which extent endogenous DVL assemblies exist (see introduction) and (ii) since recent studies addressing this issue rely on fluorescent tagging of the endogenous protein, which, among all benefits, harbors the risk to artificially affect DVL assembly. The follow-up analysis predominantly strengthens this key finding through (i) associating the detected complexes with established (DEP domain) and newly mapped (LCR4) DVL2 interaction sites, which we think is crucial to validate our biochemical approach, and (ii) linking the complexes with condensate formation and pathway activation for functional insights.

In addition, we performed new experiments with re-expression of DVL2 and our key mutants at close to endogenous expression levels in *DVL1/2/3* knockout cells, supporting a physiological relevant role of our findings (new Figs. 7G and S5K, please also see point (5) below).

(3) Mutants lacking activity/complex formation, e.g. DVL2_1-418, may need further validation. For instance, DVL2_1-506 (same mutant but with DEP) seems to form condensates and it is functional in WNT signalling (King et al., 20223). These differences could be caused by the lack of DEP domain in this particular construct and/or folding differences.

We would definitely expect that DVL2 1-506 exhibits increased condensate formation and pathway activation as compared to DVL2 1-418, since the DEP domain was thoroughly characterized as interaction domain in the Bienz lab and the Gammons lab (see references), which we confirmed in our assays (Fig. 2B-D). However, as the DEP domain is an established DVL2 interaction site, we were not interested to further characterize the DEP domain but to explain the marked difference in complex formation between DVL2 ∆DEP and 1-418 (Fig. 2A-C), which could not be associated with any known DVL2 interaction site and which we finally mapped to CFR (Fig. 4A-D).

Since fusion of the newly-characterized interaction site CFR to DVL2 1-418 (1-418+CFR) rescued complex formation, condensate formation and signaling activity (Fig. 3B-E and Fig. 4C, D), we think that the lacking activity/complex formation of DVL2 1-418 is more likely due to missing interaction sites than due to folding problems. However, as it is hard to exclude folding differences of deletion mutants, we confirmed the CFR activity through loss-of-function experiments in the context of fulllength DVL2 with minimal point mutations (Fig. 7A-G, VV,FF).

(4) The key mutants, DeltaCFR and VV/FF only show mild phenotypes. The authors' results suggest that these regions contribute but are not necessary for (1) complex formation (Density gradient Figures 7A and B), condensate formation (Figures 7C and D), and WNT activity (Figure 7E). Of note Figure 7C shows examples for the mutants with no condensates while the qualification indicates that 50% of the cells do have condensates.

Condensate formation and Wnt pathway activation by DVL VV-AA FF-AA were reduced by more than 50% as compared to WT (Fig. 7D, E). We consider these marked differences, since loss of function always ranges between 0% and 100%. In newly performed experiments in *DVL1/2/3* knockout cells, the differences were even more pronounced, see point (5) below.

Yes, Fig. 7C shows an example to qualitatively visualize the change in condensate formation, while Fig. 7D provides the corresponding quantification allowing quantitative assessment of the differences.

(5) Most of the o/e analyses (including all reporter assays) should be performed in DVL1-3 KO cells in order to explore specifically the behaviour of the investigated mutants.

As suggested, we employed *DVL1/2/3* knockout cells for performing reporter assays (T-REx *DVL1/2/3* triple knockout cells and T-REx *DVL1/2/3 RNF43 ZNRF3* penta knockout cells, which are even more sensitive towards DVL re-expression as they lack RNF43/ZNRF3-mediated degradation of DVL activating receptors; both cell lines from the Bryja lab). Here, we focused on key mutants in the context of full-length DVL2, as they are closest to the physiologic situation. Upon overexpression, DVL2 VV-AA FF-AA and DVL2 ∆CFR showed markedly reduced pathway activation as compared to WT DVL2 (new Figs. 7F and S5J). Especially in the *DVL1/2/3* triple knockout cells, DVL2 VV-AA FF-AA hardly activated the pathway and was as inactive as the established M2 mutant (new Fig. 7F). Moreover, re-expression at close to endogenous expression levels revealed that DVL2 VV-AA FF-AA less efficiently rescued Wnt3a-induced pathway activation as compared to WT (Figs. 7G and S5K).

(6) How comparable are condensates found in the cytoplasm (usually for wt DVL) with those located in the nucleus (DEP mutants)?

In principal, cytosolic condensates could differ from nuclear condensates due to various reasons, such as e.g. different protein concentration, different availability of interaction partners or different biochemical/biophysical properties (please also see point 1 of reviewer 1). In our condensatechallenging experiments (osmotic shock, 1,6-hexandiol), cytosolic condensates of full-length DVL2 and nuclear condensates of DVL2 mutants behaved quite similar (Fig. 6A-C).

We are confident that the differences between different DEP mutants in our mapping experiments are not due to nuclear localization but reflect the overall condensation capacity because later proofof-concept experiments demonstrated that CFR, which was identified in these mapping experiments, contributes to cytosolic condensate formation in the context of full-length DVL2 (Fig. 7C, D). Moreover, a new analysis focusing only on cells with cytosolic condensates, which can also be observed for DEP mutants to a low extent, revealed similar differences between key DEP mutants as observed before (Fig. S3E, F; for details please also see point 1 of reviewer 1).

Several studies in the last two decades have analysed the relevance of DVL homo - and heteroclustering by relying on overexpressed proteins. Recent studies also explored the possibility of DVL undergoing liquid-liquid phase separation following similar principles. As highlighted by the authors in the introduction, there is a need to understand DVL dynamics under endogenous/physiological conditions. Recent super-resolution studies aimed at that question by characterising endogenously edited DVL2. The authors seemed to aim in the same direction with their initial findings (Figure 1A) but quickly moved to o/e proteins without going back to the initial question. This reviewer thinks that to support their conclusions and advance in this important question, the authors should introduce the relevant mutations in the endogenous locus (e.g. by Cas9+ donor template encoding the required 3' exons, as done by others before for WNT components, including DVL2) and determine their impact in the above-indicated processes.

We agree that genomic editing of the DVL2 locus would be the cleanest system to study the relevance of CFR at endogenous expression levels. As we did not have the resources to generate the suggested cells, we, as an alternative, transiently re-expressed DVL2 and the respective mutants at low levels that were really close to the endogenous expression levels in *DVL1/2/3* triple knockout cells (Fig. S5K). These experiments revealed that DVL2 VV-AA FF-AA less efficiently rescued Wnt3ainduced pathway activation as compared to DVL2 WT (Fig. 7G).